# Boosting thermoelectric performance of single-walled carbon nanotubes-based films through rational triple treatments

Yuan-Meng Liu[1,5], Xiao-Lei Shi[2,5], Ting Wu[1], Hao Wu[1], Yuanqing Mao[2,3,4], Tianyi Cao[2], De-Zhuang Wang[1], Wei-Di Liu[2], Meng Li[2], Qingfeng Liu ✉[1] & Zhi-Gang Chen ✉[2]

Single-walled carbon nanotubes (SWCNTs)-based thermoelectric materials, valued for their flexibility, lightweight, and cost-effectiveness, show promise for wearable thermoelectric devices. However, their thermoelectric performance requires significant enhancement for practical applications. To achieve this goal, in this work, we introduce rational "triple treatments" to improve the overall performance of flexible SWCNT-based films, achieving a high power factor of 20.29 $\mu W\,cm^{-1}\,K^{-2}$ at room temperature. Ultrasonic dispersion enhances the conductivity, $NaBH_4$ treatment reduces defects and enhances the Seebeck coefficient, and cold pressing significantly densifies the SWCNT films while preserving the high Seebeck coefficient. Also, bending tests confirm structural stability and exceptional flexibility, and a six-legged flexible device demonstrates a maximum power density of 2996 $\mu W\,cm^{-2}$ at a 40 K temperature difference, showing great application potential. This advancement positions SWCNT films as promising flexible thermoelectric materials, providing insights into high-performance carbon-based thermoelectrics.

In energy utilization processes, a substantial amount of energy is dissipated in the form of heat, resulting in significant energy losses. Thermoelectric materials and devices, capable of directly converting heat into electricity and efficiently harnessing waste heat, represent a promising technology for sustainable energy[1]. To enhance the energy conversion efficiency of thermoelectric devices (TEDs), the thermoelectric materials as core components must demonstrate high thermoelectric performance[2], governed by the dimensionless figure of merit $ZT$, defined as $ZT = S^2\sigma T/\kappa^3$, where $S$, $\sigma$, $T$, and $\kappa$ represent the Seebeck coefficient, electrical conductivity, absolute temperature, and thermal conductivity, respectively[4]. Generally, achieving the decoupling of $\sigma$ and $S$, which allows an increase in both parameters simultaneously, poses a significant challenge in thermoelectric

materials[5]. Additionally, the most appropriate strategy for enhancing the thermoelectric performance of materials may effectively reduce the $\kappa$ of materials by delicately tailoring their micro/nanostructures and reasonably assembling them into well-defined macroscopic configuration[5].

Currently, inorganic thermoelectric materials are considered to exhibit high thermoelectric performance due to their controllable high $\sigma$ and $S$[6]. However, inorganic materials often suffer from major drawbacks such as high cost[5], rigidity[5], and toxicity[5], restricting their practical applications, particularly in the domain of wearable flexible TEDs (F-TEDs)[7]. In stark contrast, most organic materials (e.g., conducting polymers) inherently possess lower $\kappa$[8]. However, owing to their lower $S$ and challenges in effective enhancement[8], the power factor $S^2\sigma$ is low[8],

[1]State Key Laboratory of Materials-Oriented Chemical Engineering, College of Chemical Engineering, Nanjing Tech University, Nanjing, China. [2]School of Chemistry and Physics, ARC Research Hub in Zero-emission Power Generation for Carbon Neutrality, and Centre for Materials Science, Queensland University of Technology, Brisbane, QLD, Australia. [3]School of Mechanical and Mining Engineering, The University of Queensland, Brisbane, QLD, Australia. [4]Department of Physics and Guangdong Provincial Key Laboratory of Computational Science and Material Design, Southern University of Science and Technology, Shenzhen, China. [5]These authors contributed equally: Yuan-Meng Liu, Xiao-Lei Shi. ✉e-mail: qfliu@njtech.edu.cn; zhigang.chen@qut.edu.au

resulting in significantly inferior thermoelectric performance compared to inorganic materials. Carbon-based thermoelectric materials have attracted considerable research interest due to their flexibility, non-toxicity, low cost, and ease of processing[9]. One-dimensional (1D) single-walled carbon nanotubes (SWCNTs), exhibit high $\sigma$, and unique electronic structures[9], rendering them capable of achieving a relatively high $S$[9–11]. However, besides the high thermal conductivity of SWCNTs, due to the strong van der Waals forces and a large aspect ratio (>1000) of SWCNTs, SWCNTs tend to form large bundles during processing, posing challenges to dispersion and significantly limiting the realization of their excellent thermoelectric performance in practical applications.

Conventional methods for dispersing SWCNTs in various solutions include high-speed shearing, ultrasonic treatment, non-covalent functionalization, and covalent functionalization[12–14]. For example, high-frequency ultrasonication and ball milling, rely on substantial mechanical force to facilitate the dispersion of SWCNTs[12–14]. Some studies have explored the use of specific surfactants or polymers to enhance the dispersion of SWCNTs in solution, often accompanied by changes in the purity and diameter distribution of the SWCNTs during the dispersion process[15–18]. For instance, the use of different sorting polymers can adjust the purity of SWCNTs from 94% to 99% and the average diameter of SWCNTs from 1.0 to 1.2 nm, resulting in an approximately 10-fold increase in the $S^2\sigma$, reaching 2.43 μW cm$^{-1}$ K$^{-2}$ [19].

In addition to addressing the challenges of dispersion, various strategies have been proposed to enhance the thermoelectric performance of SWCNTs, including chiral sorting, chemical doping, and organic composites[9,20–22]. Current research primarily focuses on achieving high thermoelectric performance in SWCNTs through the sorting of metallic and semiconducting types (denoted as M-SWCNTs and S-SWCNTs, respectively). The distinct band structures of M- and S-SWCNTs allow for the coordination of the $S$ and $\sigma$ through band tuning[23,24]. Researchers have explored the dependency of the $S$, $\sigma$, and $S^2\sigma$ on the Fermi-level position in SWCNT films with varying M-SWCNTs content[25]. It was observed that high-purity aligned M-SWCNT films (purity > 99%) exhibited an $S^2\sigma$ approximately five times higher than that of highest-purity single-chirality (6,5) S-SWCNT films, reaching around 3 μW cm$^{-1}$ K$^{-2}$. The superior thermoelectric performance of M-SWCNTs is mainly attributed to the simultaneous enhancement of the 1D conduction electron-based $S$ and $\sigma$ near the first Van Hove singularity. However, the cost of chiral sorting is high, making it challenging to re-sort already produced SWCNTs. Additionally, chemical doping is a common method for adjusting the thermoelectric performance of SWCNTs, utilizing solvents such as acids, bases, and organic small molecules. The doping process involves injecting charge carriers into SWCNTs or inducing Coulomb interaction on the SWCNT surface[9,26–28]. Studies indicate that the effects of acid-base treatment of SWCNTs are unstable and may potentially damage the intact tubular structures. Single-solvent treatment has a limited impact on improving the thermoelectric performance of SWCNTs. For example, researchers demonstrated an improvement in charge carrier transport in enriched semiconductor SWCNT networks through chemical injection[29]. Using functionalized icosahedral boron clusters as dopants can reduce Coulomb interactions between holes and accompanying counterions, ultimately achieving a high $S^2\sigma$ value, ~9.17 μW cm$^{-1}$ K$^{-2}$. Researchers have also enhanced thermoelectric performance through composite materials, combining SWCNTs with organic thermoelectric materials[9,20–22]. The energy barrier formed between SWCNTs and polymers induces an energy filtering effect, leading to a potentially high $S$. This approach has been widely accepted and applied. The low $\kappa$ of polymers effectively reduces the overall $\kappa$ of SWCNT/polymer hybrids. The most widely used polymers for composites are poly(3,4-ethylenedioxythiophene):poly(styrenesulfonate) (PEDOT:PSS)[30–34] and polyaniline (PANI)[35,36]. Some work achieved high thermoelectric performance in PEDOT:PSS/SWCNTs composite films by combining them in a specific ratio and using dimethyl sulfoxide doping and NaBH$_4$ dedoping[37]. By simultaneously improving $\sigma$ (1718 S cm$^{-1}$) and $S$ (49 μV K$^{-1}$), the PEDOT:PSS/SWCNTs composite film exhibited a maximum $S^2\sigma$ of 4.11 μW cm$^{-1}$ K$^{-2}$. However, although various strategies effectively enhance the thermoelectric performance of SWCNTs, there remains significant room for improvement. Currently, reported $S^2\sigma$ values for SWCNT films are mostly below 10 μW cm$^{-1}$ K$^{-2}$ [38].

In this work, we sought to explore a comprehensive approach involving multiple effective treatments to achieve a high and stable $S^2\sigma$ value for pure SWCNT films, as depicted in the triple treatment optimization scheme presented in Fig. 1a. This study introduces a pure SWCNT film with a record-high $S^2\sigma$ value of 20.29 μW cm$^{-1}$ K$^{-2}$. The undoped and uncompounded p-type SWCNTs were optimized by adjusting ultrasonic time to achieve favorable dispersion and subsequently assembled into a thin film by solution casting. Then, through NaBH$_4$ solvent treatment, the $S$ of the SWCNT film was maximally increased to approximately 42 μV K$^{-1}$. Subsequently, a cold-pressing process was applied to densify the microstructure of the SWCNT film. Because of densification, the tight connection between nanotubes in the SWCNTs film was strengthened, resulting in a substantial increase in $\sigma$ while almost maintaining the $S$. The significant enhancement in $\sigma$ complemented the sacrifice made during NaBH$_4$ solvent treatment, ultimately reaching 14,500 S cm$^{-1}$. This work showcases the highest $S^2\sigma$ value among the chiral-sorted or organically compounded SWCNT films reported so far, as illustrated in Fig. 1b (specific values provided in Supplementary Table 1)[19,29,30,32,35–37,39–42]. Bend tests demonstrate that the highly thermoelectric SWCNT film retains excellent flexibility, with

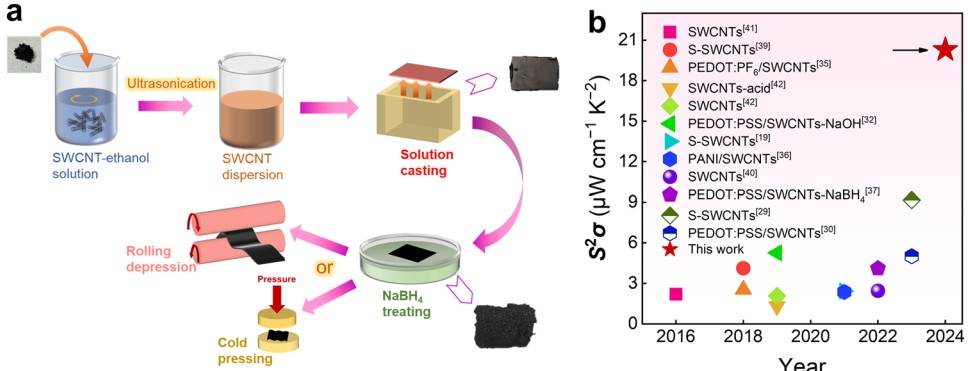

**Fig. 1 | Introduction of fabricating single-walled carbon nanotube (SWCNT) film with high thermoelectric performance by a simple triple treatment process.** **a** Schematic diagram of fabricating SWCNT film by a simple triple treatment process. **b** Comparison of power factor $S^2\sigma$ between this work and reported literature based on SWCNTs[19,29,30,32,35–37,39–42]. S-SWCNTs semiconducting SWCNTs, PEDOT:PF6 poly(3,4-ethylenedioxythiophene):hexafluorophosphate, PEDOT:PSS poly(3,4-ethylenedioxythiophene):poly(styrenesulfonate), PANI polyaniline.

a resistance error of only within 5% after 2000 bends, making it conducive to widespread utilization. Furthermore, a six-legged device fabricated using this film can generate an open-circuit voltage $V_{oc}$ of 9.8 mV at a temperature difference $\Delta T$ of 40 K, yielding a maximum output power $P$ of 0.86 μW and a maximum power density of 2996 μW cm$^{-2}$. Consequently, this process innovation charts a new course for the design of high-performance carbon-based thermoelectric materials.

## Results

The process for fabricating SWCNT films is depicted in Fig. 1a. The preparation of SWCNT films mainly involves three steps: ultrasonic dispersion of SWCNTs in ethanol, post-treatment with a NaBH$_4$ solution, and cold pressing/rolling. Film formation is accomplished using a mold-casting method, as opposed to the commonly employed vacuum filtration approach[43]. This decision is motivated by the ease of ethanol removal, the solvent used for dispersion, and the simplicity and convenience associated with mold casting. Films prepared through mold casting are more easily peeled off and separated than those produced through vacuum filtration. Additionally, under natural air drying, the compression of SWCNT films prepared by mold casting is minimized, resulting in enhanced $\sigma$ and the surface of the SWCNT films exhibit a smooth and uniform appearance.

### Ultrasonic treatment

Figure 2a–c compares the room temperature thermoelectric properties ($\sigma$, $S$, and $S^2\sigma$) of SWCNT films as a function of ultrasonication time. In the absence of ultrasonication, SWCNTs fail to achieve uniform dispersion in the mold after drying, resulting in an incomplete film formation (Supplementary Fig. 1). Therefore, ultrasonication commences with a minimum time of 5 min. As illustrated in Fig. 2a, with increasing ultrasonication time, the $\sigma$ of the SWCNT film generally follows a pattern of initial increase followed by a subsequent decrease. The $\sigma$ peaks at 3584 S cm$^{-1}$ when the ultrasonication time is 10 min. Further increases in ultrasonication time lead to a significant decline in $\sigma$, reaching a minimum of ~2100 S cm$^{-1}$ in 30 min. As the ultrasonication time extends to 40–45 min, the $\sigma$ experiences a slight increase but remains below that at 10 min. This trend may be primarily attributed to the optimal interconnection of SWCNTs under a specific ultrasonication time, resulting in the highest $\sigma$ of the SWCNT film. Additional

ultrasonication time disrupts the initial unabridged microstructures of SWCNTs, leading to a decline in $\sigma$. Especially, after a long ultrasonication time, SWCNTs may undergo breakage due to prolonged ultrasonic shearing forces, with the tight interconnection between SWCNTs slightly boosting the $\sigma$. Throughout the entire ultrasonication time, the $S$ of SWCNTs fluctuates only within a narrow range, around 2–3 μV K$^{-1}$ (Fig. 2b). Consequently, with an ultrasonication time of 10 min, the $S^2\sigma$ value reaches its optimized value, approximately 1.88 μW cm$^{-1}$ K$^{-2}$ (Fig. 2c).

To elucidate the impact of ultrasonication time on the arrangements of SWCNTs, SEM characterization was conducted on SWCNT films. As illustrated in Fig. 2d–f, SWCNTs intricately intertwine and connect with each other, forming an extensive conductive network on the film surface. With an increase in ultrasonication time, the initially disordered distribution of SWCNT threads gradually transforms into bundles with a certain directional alignment. At 10 min of ultrasonication, SWCNTs predominantly form bundles interspersed with small portions of individual threads, aligning in the comparable direction, which facilitates electron transport and subsequently enhances the $\sigma$. As ultrasonication time continues to increase, reaching 35 min, SWCNT threads completely transition into bundles, and the surface exhibits larger clusters. The initial orderly arrangement becomes more chaotic and tangled, resulting in a reduction in $\sigma$. The low- and high-magnification SEM images of SWCNT films by different ultrasonic times are provided in Supplementary Fig. 2–3 for reference. By applying ultrasonication, the concentration and arrangement of SWCNTs were altered, leading to changes in conductivity. With a certain duration of ultrasonication (10 min), the original aggregated state of SWCNTs reached a stable bundled state. However, with further increasing the duration of ultrasonication, the arrangement of SWCNTs shifted from ordered back to disordered. Therefore, after reaching optimal performance, prolonged exposure to ultrasonication will lead to a decrease in the thermoelectric properties of SWCNT films.

### NaBH$_4$ treatment

Figure 3a–c shows the impact of different NaBH$_4$ concentrations on the thermoelectric performance and structures of SWCNT films in a well-dispersed state. With increasing the concentration of NaBH$_4$ solution, the $S$ of the film initially rises before decreasing, while the $\sigma$

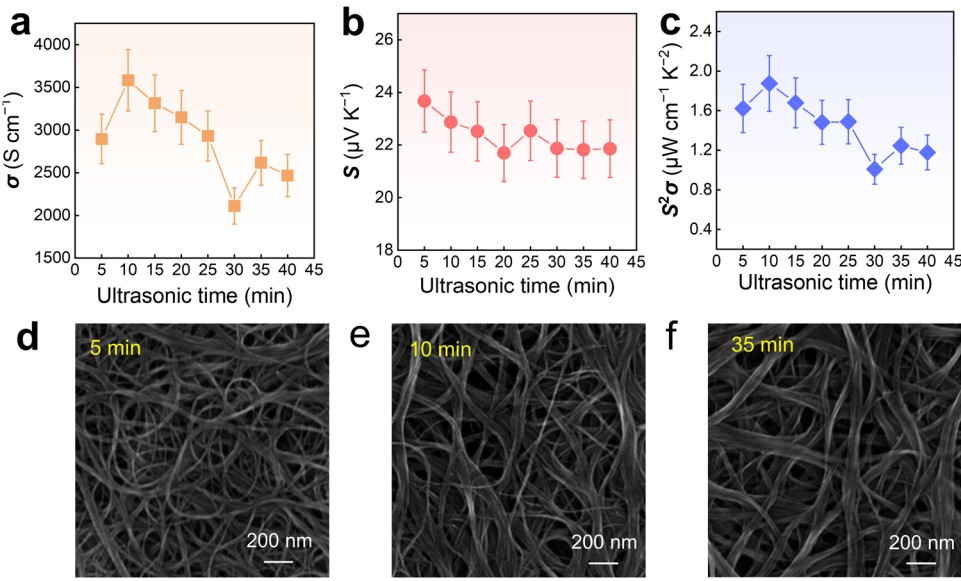

**Fig. 2 | Thermoelectric performance and interconnected networks of SWCNT films with different ultrasonic durations. a** Electrical conductivity $\sigma$, **b** Seebeck coefficient $S$, and **c** $S^2\sigma$ of SWCNT films as a function of ultrasonic time. Scanning electron microscopy (SEM) images of SWCNT films by ultrasonic times of **d** 5 min, **e** 10 min, and **f** 35 min.

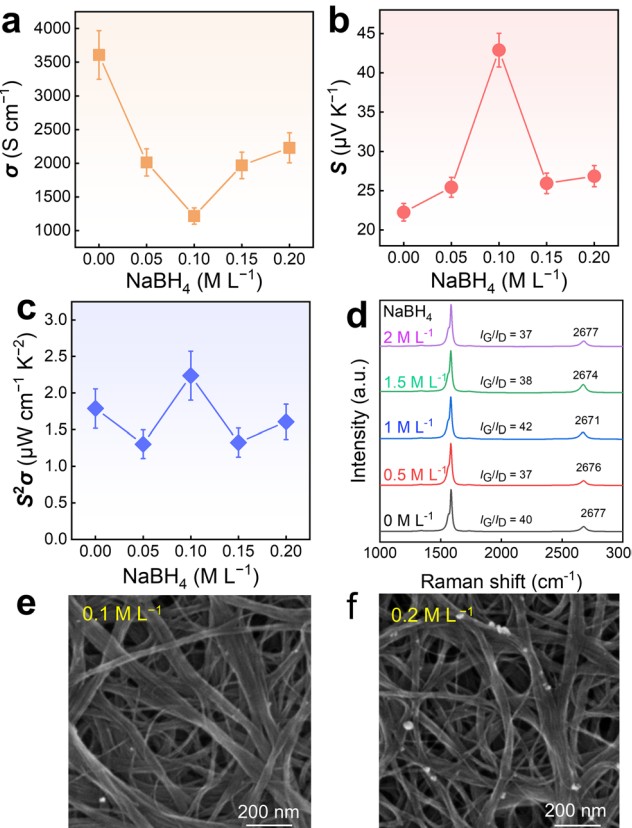

**Fig. 3 | Thermoelectric performance and interconnected networks of SWCNT films treated with different concentrations of NaBH4.** **a** $\sigma$, **b** $S$, and **c** $S^2\sigma$ of SWCNT films as a function of NaBH$_4$ concentration. **d** Raman spectra of SWCNT films treated with different concentrations of NaBH$_4$. SEM images of SWCNT films with **e** 0.1 M L$^{-1}$ and **f** 0.2 M L$^{-1}$ of NaBH$_4$.

exhibits an inverse trend, decreasing first and then increasing. This phenomenon is attributed to the harmonic relationship between the $S$ and $\sigma$, resulting in opposite trends as the carrier density changes[44,45]. At a NaBH$_4$ concentration of 0.1 M L$^{-1}$, the $S$ of the treated SWCNT film increases to 42 µV K$^{-1}$, indicating a positive effect on the film performance from the perspective of the $S^2\sigma$ value. During the NaBH$_4$ treatment process, vigorous reactions occur on the film surface, generating numerous bubbles, which is a phenomenon consistent with the literature[46]. However, the performance changes differ from the n-type doping results reported in the literature. This discrepancy could be attributed to the varying degrees of influence on the film under low and high NaBH$_4$ concentrations. At higher concentrations, there was no significant improvement in the properties of SWCNT films, as depicted in Supplementary Fig. 4. Additionally, film delamination occurs during the treatment process, transforming the film from a smooth and compact state to a rough and loose structure, accompanied by an increase in thickness, as illustrated in the SEM image of the film after NaBH$_4$ treatment in Fig. 1a.

We analyzed the mechanism of different NaBH$_4$ concentration treatments on SWCNT films by Raman characterization, as shown in Fig. 3d. In the SWCNT film, the very small disorder-induced D peaks at ~1338 cm$^{-1}$ represent the high quality of the SWCNTs, a strong G peak at 1583 cm$^{-1}$ signifies the in-plane vibration of the $sp^2$ carbon atoms, and there is a smaller second-order harmonic 2D peak around 2677 cm$^{-1}$. The intensity ratio of the G/D peaks ($I_G/I_D$) reflects the defect level of the SWCNTs, while the intensity of the 2D peak indicates the layering phenomenon in the film[47–49]. There is no shift in the G peak before and after NaBH$_4$ treatment, indicating that the doping level has no impact on the 1D structure of the SWCNTs. The original SWCNT film

has a relatively high G/D peak intensity ratio, approximately 40, indicating high purity with fewer defects. With an increase in treatment concentration, the intensity ratio of G/D peaks first decreases, then increases, and finally decreases again. This overall trend aligns with the trend of $S^2\sigma$. When the NaBH$_4$ treatment concentration is 0.1 M L$^{-1}$, the highest $I_G/I_D$ is 42, suggesting that a certain concentration of NaBH$_4$ is somewhat beneficial for reducing surface defects of SWCNTs, thereby increasing the $S$ of the SWCNT film. Generally, defects typically lead to electron localization and changes in the band structure, which is one of the significant reasons influencing the $S$[46,50,51]. With increasing the concentration of NaBH$_4$ treatment, the intensity of the 2D peak was generally increased first and then decreased. This is mainly attributed to the varying effects of interlayer separation among films treated with different concentrations of NaBH$_4$. The frequency of radial breathing mode (RBM) serves as a distinctive phonon mode inversely proportional to the SWCNT diameter, and we confirmed the SWCNT nature of the film through its observation in the Raman spectra. The RBM peaks clearly illustrate the highly stable tubular structures of SWCNTs in these films after the treatments of NaBH$_4$ at various concentrations, as depicted in Supplementary Fig. 5. Figure 3e, f shows SEM characterizations of the surfaces of the SWCNT films treated with 0.1 M L$^{-1}$ and 0.2 M L$^{-1}$ of NaBH$_4$. It reveals a large and disorderly distribution of SWCNT bundles after treatment with NaBH$_4$. This suggests that during NaBH$_4$ treatment, the film becomes rough, disrupting the distribution on the SWCNT surface, and reducing the connection between layers and SWCNTs, therefore leading to a decrease in $\sigma$. Moreover, it is evident that with an increase in NaBH$_4$ concentration, the looseness between SWCNTs increases, and some residuals remain on the surface of SWCNTs (Fig. 3f and Supplementary Fig. 6). In addition, we conducted Hall measurements on the film samples (Supplementary Fig. 7a). The results indicate that with increasing the NaBH$_4$ concentration, the hole carrier concentration $n_p$ of the film initially decreased and then leveled off. The $S$ increased as the $n_p$ decreased, consistent with experimental data. Furthermore, the carrier mobility $\mu$ of the film increased with increasing the NaBH$_4$ concentration. This is attributed to the weakening of carrier scattering among carriers due to the decrease in $n_p$.

To theoretically demonstrate the variation in the thermoelectric properties of SWCNT films, we conducted first-principles density functional theory (DFT) calculations on SWCNTs with different chiral structures. Generally, SWCNTs exhibit three chiral structures[9]. Figure 4a–c illustrates the calculated band structures of pristine SWCNTs with chiral indices $(2n + m)$ mod 3 = 0, 1, and 2, respectively. From the calculations, it is evident that SWCNTs with chiral index $(2n + m)$ mod 3 = 0 overlap in the conduction and valence bands, exhibiting metallic properties with almost no bandgap, thus demonstrating electron conduction characteristics (n-type). On the other hand, SWCNTs with chiral indices $(2n + m)$ mod 3 = 1 and 2 exhibit indirect bandgaps >0.6, indicating semiconductor characteristics. Since the SWCNTs used in the experiment are p-type as indicated by the measured positive $S$, our SWCNTs exhibit semiconductor behaviors, specifically $(2n + m)$ mod 3 = 1 or 2. Figure 4d–f shows the calculated band structures of SWCNTs with a single carbon vacancy defect for chiral indices $(2n + m)$ mod 3 = 0, 1, and 2, respectively. It can be observed that upon removing one carbon atom to form a vacancy defect, the bandgaps of SWCNTs with $(2n + m)$ mod 3 = 1 or 2 decrease, enhancing carrier transition capabilities and in turn leading to an increase in $\sigma$ and a decrease in the $S$. Considering that after treatment with a certain concentration of NaBH$_4$ in the experiment, the defects in SWCNT films decrease, resulting in an increase in the bandgap and $S$, which is consistent with experimental results. We also calculated the charge distribution functions and electron localization functions before and after the formation of a vacancy defect, as shown in Fig. 4g–j. For p-type SWCNTs, with an increase in defect content, the carrier density around the defect increases. Therefore, after treatment

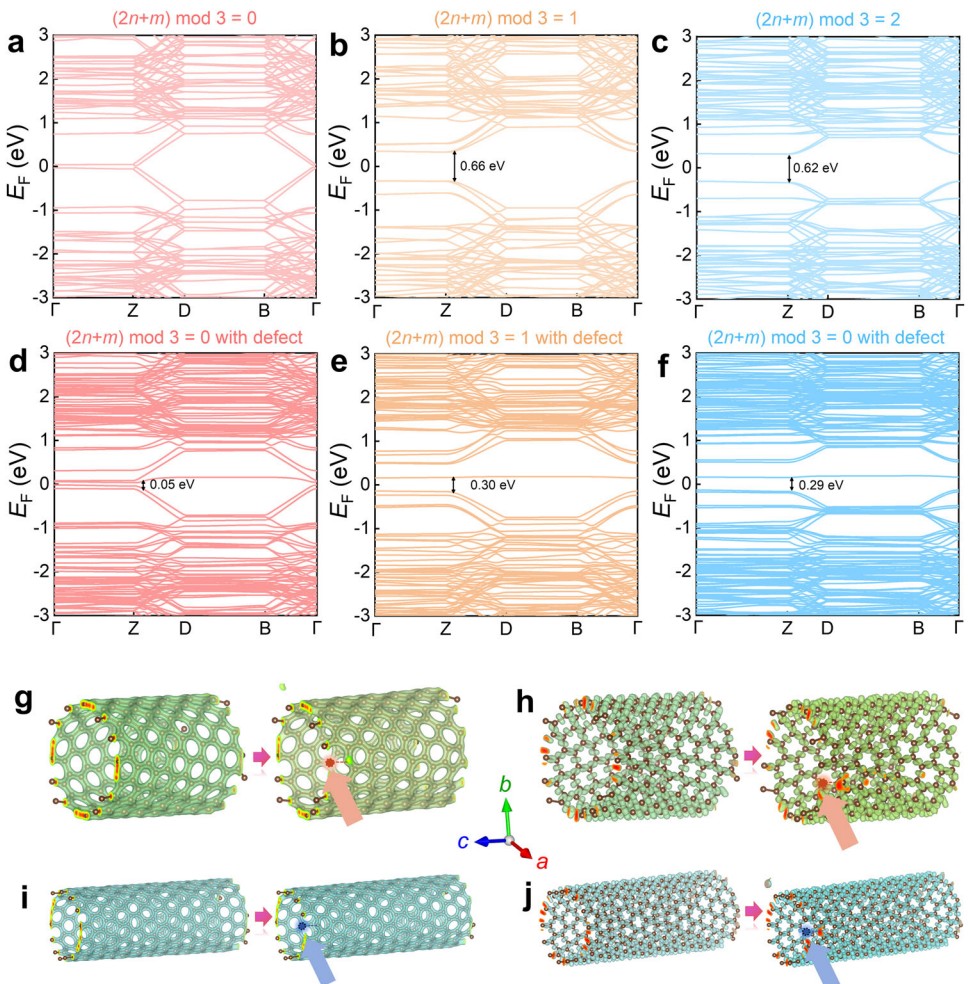

**Fig. 4 | Calculation results of SWCNT films at different (n, m) indices with and without defects.** Calculated band structures of pristine SWCNTs for **a** type $(2n + m)$ mod 3 = 0, **b** type $(2n + m)$ mod 3 = 1, and **c** type $(2n + m)$ mod 3 = 2; and calculated band structures of SWCNTs with a carbon vacancy for **d** type $(2n + m)$ mod 3 = 0, **e** type $(2n + m)$ mod 3 = 1, and **f** type $(2n + m)$ mod 3 = 2. **g** Charge density distribution diagram and **h** electron localization function diagram of SWCNTs for type $(2n + m)$ mod 3 = 1 with and without carbon vacancy. **i** Charge density distribution diagram and **j** electron localization function diagram of SWCNTs for type $(2n + m)$ mod 3 = 2 with and without carbon vacancy. The positions with carbon vacancies are indicated with arrows.

with 0.1 M L$^{-1}$ NaBH$_4$, the defects in SWCNT films decrease, leading to a decrease in $n_p$ and $\sigma$.

### Cold pressing/rolling treatment

To further enhance the $S$, the SWCNT films were subjected to rolling/cold pressing. Figure 5 explores the impact of cold pressing and rolling on the thermoelectric performance and structures of SWCNT films. As seen in Fig. 5a, both the cold-pressed and rolled SWCNT films exhibit a significant increase in $\sigma$, caused by the densified structure of the films[52–55]. The $\sigma$ is even higher after cold-pressing, but the increase in $\sigma$ after cold-pressing/rolling for the untreated SWCNT films is only slightly higher than that for NaBH$_4$-treated SWCNT films. This is because NaBH$_4$-treated SWCNT films have a smaller initial $\sigma$, and the degree of compression is the same for different films during cold-pressing/rolling. The $\sigma$ of the SWCNTs-NaBH$_4$ film increases to 14,300 S cm$^{-1}$ after cold-pressing, nearly 12 times the original value, while it increases to 11,000 S cm$^{-1}$ after rolling, about nine times the original value. Figure 5b shows that the $S$ of the SWCNT film after cold-pressing/rolling only slightly decreases. The $S$ of the SWCNTs-NaBH$_4$ film (after NaBH$_4$ treatment) decreases from 42 to 37 μV K$^{-1}$ after cold-pressing and to 40 μV K$^{-1}$ after rolling. Since the NaBH$_4$-treated film has a higher $S$ compared to the original counterpart, the NaBH$_4$-treated SWCNT film exhibits a higher $S^2\sigma$ value after cold-

pressing/rolling, as shown in Fig. 5c. The $S^2\sigma$ values for the rolled NaBH$_4$-SWCNT film and the cold-pressing NaBH$_4$-SWCNT film are approximately 17.87 μW cm$^{-1}$ K$^{-2}$ and 20.29 μW cm$^{-1}$ K$^{-2}$, respectively, indicating both methods are effective for boosting the thermoelectric performance of SWCNT films. To further understand the thermoelectric performance of as-fabricated SWCNT films, the in-plane $\kappa$ of the 0.1 M L$^{-1}$ NaBH$_4$-treated SWCNT film after cold pressing was measured to be ~676 W m$^{-1}$ K$^{-1}$. This value is much lower than the reported theoretical $\kappa$ of SWCNTs, (e.g., 1000–3000 W m$^{-1}$ K$^{-1}$)[56], which can be attributed to the well-interconnected and densified SWCNT networks with uniform thickness. The interfaces between the SWCNTs determine the potential for further enhancing the phonon scattering, and in turn, reducing lattice thermal conductivity $\kappa_l$. However, after triple treatments, there is a slight increase in the $\kappa$ of SWCNT films, but the final $ZT$ value remains significantly higher than that of the pristine film (specific values provided in Supplementary Fig. 8). The increase in $\kappa$ of SWCNT films after pressing is attributed to the densification of the initially porous film structure, facilitating heat transfer. However, as the electrical transport properties become superior after pressing, offsetting the adverse effects of increased $\kappa$, the final $ZT$ value is enhanced. The determined $ZT$ value of our fabricated SWCNT films is ~0.001, which is more than sufficient for serving as low-grade thermal harvesters and temperature sensors.

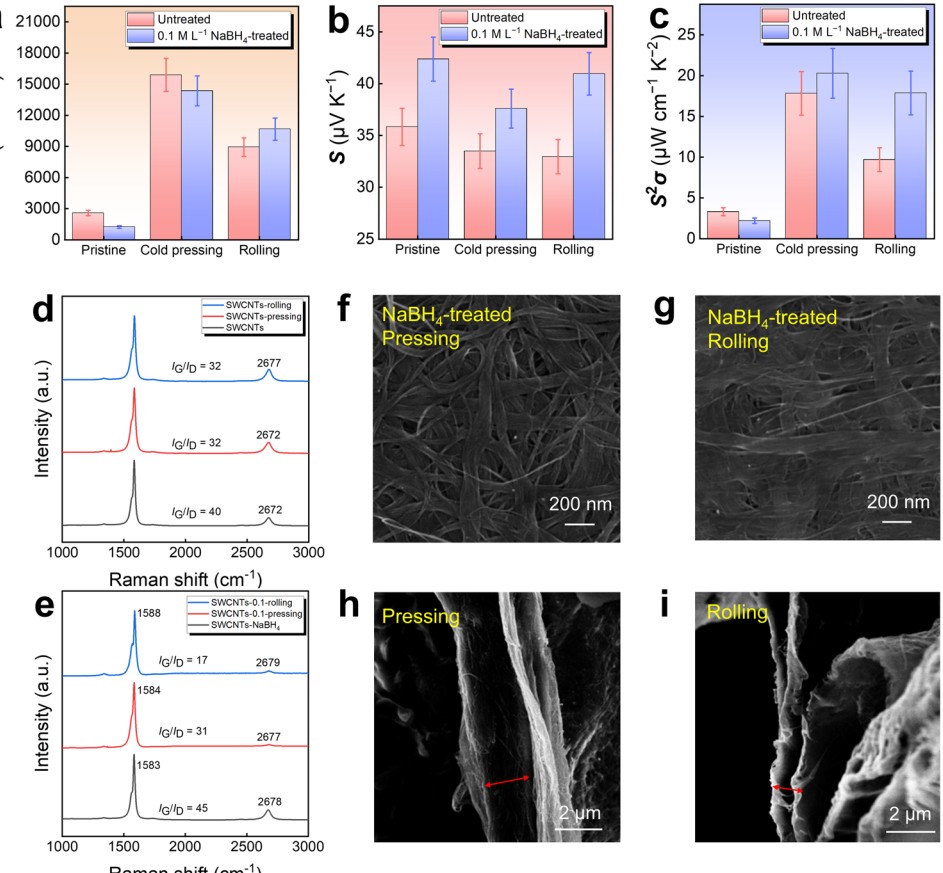

**Fig. 5 | Impact of cold pressing and rolling on the thermoelectric performance and interconnected networks of SWCNT films.** **a** $\sigma$, **b** $S$, and **c** $S^2\sigma$ of pristine, cold-pressed, and rolled SWCNT films with and without 0.1 M L$^{-1}$ NaBH$_4$ treatment. Raman spectra of **d** the untreated and **e** 0.1 M L$^{-1}$ NaBH$_4$-treated pristine, cold-pressed, and rolled SWCNT films. SEM images of 0.1 M L$^{-1}$ NaBH$_4$-treated SWCNT films after **f** cold pressing and **g** rolling from top views, and **h**, **i** corresponding SEM images from cross-sectional views.

Figure 5d, e shows the Raman spectra of untreated and 0.1 M L$^{-1}$ NaBH$_4$-treated pristine, cold-pressed, and rolled SWCNT films. After cold-pressing, the G peak of both the pristine SWCNT film and the NaBH$_4$-treated SWCNT film remains unchanged, indicating that their doping levels remain constant, as shown in Fig. 5d, e. However, the G/D peak ratio decreases, suggesting that the cold-pressing process causes the breaking of some bonds in SWCNTs, leading to an increase in defects[52–55]. Additionally, the intensity of the 2D peak was decreased after cold-pressing, with ratios of 1:0.63 and 1:0.25, indicating a tighter connection between SWCNT layers, strengthening the longitudinal transport of electrons between SWCNT layers, and significantly increasing the $\sigma$[49]. On the other hand, the Raman characterization slightly differs for the original SWCNT film and the NaBH$_4$-treated SWCNT film after rolling. The G peak position of the original SWCNT film remains unchanged, and the G/D peak ratio decreases, indicating an increase in defects. However, after rolling, the G peak of the NaBH$_4$-treated SWCNT film undergoes a blue shift, and the G/D peak ratio decreases even more. This suggests that the lateral pressure generated during rolling causes stretching and deformation in the layered SWCNT film[57]. The structural changes in SWCNTs are more pronounced because the interlayer gap in pure SWCNT films is smaller, resulting in less influence from the lateral tension during rolling. Moreover, the intensity of the 2D peak remains relatively constant after rolling for pure SWCNT films, whereas for the treated SWCNT film, the 2D peak intensity undergoes a significant decrease, with ratios of 1:1.08 and 1:0.055. This suggests that rolling has a pronounced effect on the close connection between the layers of the treated SWCNT film after rolling. Changes in the RBM intensity also reveal the impact of cold pressing

and rolling, as illustrated in Supplementary Fig. 9. The RBM intensity was decreased in SWCNT films following cold pressing or rolling, attributed to increased defects induced by these processes, which weaken the radial vibration of SWCNTs. However, a notable distinction is observed, wherein the RBM peak of the NaBH$_4$-SWCNT film with rolling nearly disappears, indicating that the lateral pressure from rolling induces substantial structural damage to the SWCNTs[57]. Simultaneously, theoretical calculations also indicate that the introduction of defects on SWCNTs leads to a reduction in the bandgap of SWCNT films, resulting in an increase in $n_p$, and in turn, a decrease in $S$ and an increase in $\sigma$, as shown in Fig. 4. Therefore, the potential generation of defects during the cold-pressing process is also a reason for the change in the thermoelectric properties of SWCNT films. Although the G/D peak ratio of SWCNT films decreases slightly after cold pressing, indicating a slight increase in defects and an enhancement in the $\sigma$ of the film, the decrease in the G/D value is small, suggesting that defects have little effect on the enhancement of $\sigma$. However, after cold pressing, the connections between SWCNTs in the film become tighter, which significantly improves the $\sigma$ of the film. From the intensity of the 2D peak, it can be observed that while maintaining the SWCNT structure, both the 2D peak ratio and the thickness of the film significantly decrease. After the film thickness decreases significantly, the overall volume of the film decreases substantially. However, since the change in the total carrier quantity is not significant (due to a relatively constant defect concentration), the sharp increase in $n_p$ is primarily due to the significant decrease in the overall volume of the film (Supplementary Fig. 7b). Therefore, the densification of SWCNT films after cold pressing plays a dominant role in the high conductivity of SWCNT films.

Figure 5f, g shows top-view SEM images of 0.1 M L$^{-1}$ NaBH$_4$-treated SWCNT films after cold pressing and rolling, and Fig. 5h, i shows their cross-sectional SEM images. More SEM results are provided in Supplementary Fig. 10–12 for reference. It is evident that after cold-pressing, the SWCNT films were densified with a decreased thickness under vertical pressure, leading to interlayer stacking and overlapping. After rolling, the generated lateral pressure causes the SWCNT films to become flatter and elongated. Since the NaBH$_4$-treated film is rather looser, the stretching effect of rolling becomes more pronounced, as illustrated in Supplementary Fig. 11. From the cross-section SEM images, it is observed that the film thickness decreases to approximately 2 μm after cold-pressing. By tearing through the film layers, the thickness can further decrease to around 1.2 μm. After rolling, the film thickness reduces to about 1.8 μm. The thickness of the pure SWCNT film is relatively uniform at 8 μm, but after treatment, the film exhibits layering with larger interlayer gaps. After multiple measurements, the post-treatment film thickness is approximately 12 μm, as shown in Supplementary Fig. 12. The treatments of rolling and cold-pressing result in the densification of the SWCNT films with a smoother surface, decreased film porosity, and increased density, and in turn a substantial improvement in σ, as illustrated in Supplementary Fig. 13.

To analyze the elemental composition and valency states of the SWCNT films, X-ray photoemission spectroscopic (XPS) spectra were employed for untreated and 0.1 M L$^{-1}$ NaBH$_4$-treated pristine, cold-pressed, and rolled SWCNT films, as illustrated in Fig. 6a–c. The C1s

(284.8 eV) and O1s (531.9 eV) peaks indicate that carbon is the primary element in SWCNT films, with a trace amount of oxygen, likely absorbed from the air during exposure (specific values provided in Supplementary Tables 2 and 3). Charge correction for C1s and O1s peaks was performed against untreated SWCNTs. Following NaBH$_4$ treatment, no additional elements such as B or Na are introduced, suggesting that NaBH$_4$ solvent treatment does not adsorb onto the surface of SWCNTs but rather induces valence state and defect regulation, as shown in Supplementary Fig. 14. In Fig. 6b, the C1s peak shifts towards higher energy after NaBH$_4$ treatment, indicating an enhancement in the oxidation state of the carbon peak within the carbon tube. Figure 6c reveals that carbon in the SWCNT films primarily exists in the form of C–C bonds, with a small amount combining with oxygen to form C–O bonds and C=O bonds. Moreover, cold pressing and rolling exhibit minimal impact on the valence states of C and O bonds in the SWCNTs[58–61].

To further explore the structural and stability changes in SWCNT films pre- and post-cold pressing treatment, we subjected the films to specific ultrasonic treatment durations (30 min), comparing samples with and without cold pressing. Subsequently, we deposited these ultrasonically treated SWCNTs onto a copper grid, which were then examined using transmission electron microscopy (TEM). Figure 6d–f displays TEM images of films without cold pressing treatment. In Fig. 6d, the film surface morphology exhibits a loose structure after ultrasonic treatment, making it challenging to identify the presence of

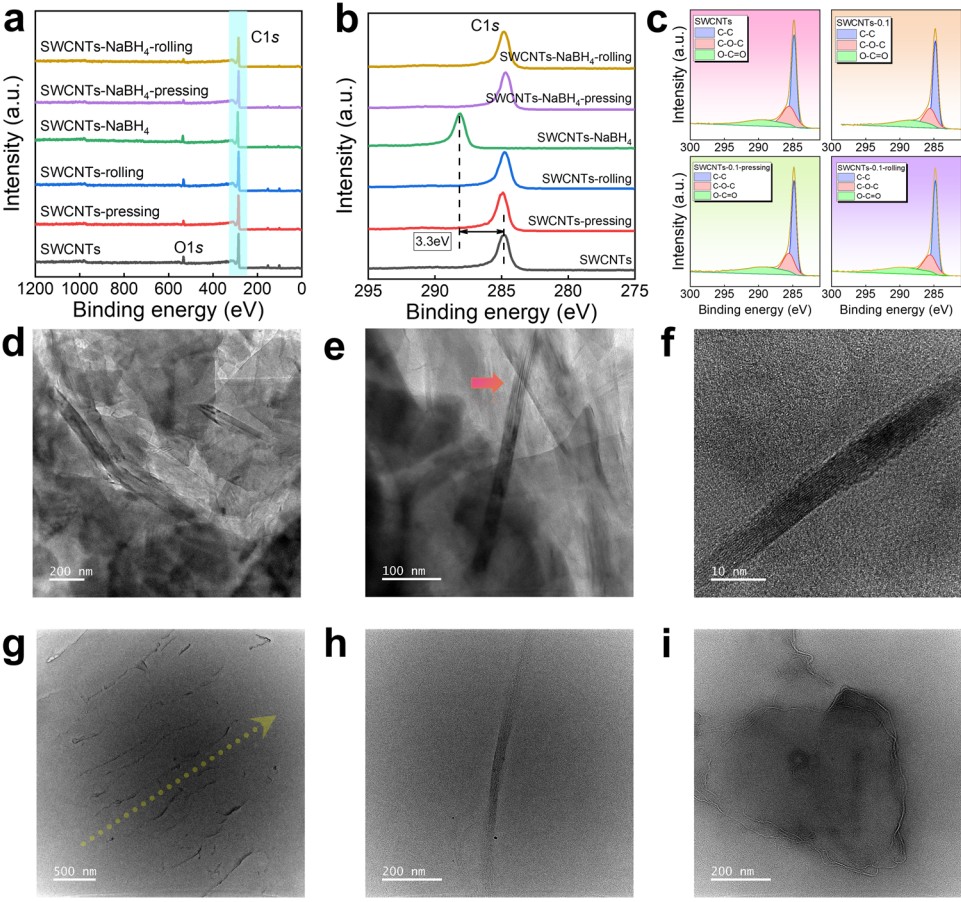

**Fig. 6 | Characterizations of structures, atom binding information, and structures of SWCNT films. a** X-ray photoelectron spectroscopy (XPS) pattern with a broad range, and **b** corresponding magnified C1s patterns of pristine, cold-pressed, and rolled SWCNT films with and without 0.1 M L$^{-1}$ NaBH$_4$ treatment. **c** Specific C1s multi-peak patterns of pristine, 0.1 M L$^{-1}$ NaBH$_4$ treatment, cold-pressed, and rolled SWCNT films with 0.1 M L$^{-1}$ NaBH$_4$ treatment. **d** Transmission electron microscopy (TEM) images of 0.1 M L$^{-1}$ NaBH$_4$-treated SWCNT film before cold pressing. **e** Corresponding magnified TEM image. The arrow indicates one potential SWCNT. **f** High-resolution TEM (HRTEM) image of one separated SWCNT from the film. **g** TEM images of 0.1 M L$^{-1}$ NaBH$_4$-treated SWCNT film after cold pressing. The arrow indicates the potential orientation. **h, i** Corresponding magnified TEM images to show the interfaces within the film.

SWCNTs and suggest potential damage to SWCNTs by ultrasonication[62]. Figure 6e shows an image of an undamaged SWCNT with rough lattice information found after ultrasonication, confirmed by its tube diameter. This indicates that a majority of SWCNTs were damaged after ultrasonic treatment, leaving only a small fraction maintaining their tubular morphology. Figure 6f presents a single SWCNT separated from the film structure, suggesting low structural stability of the film without cold pressing due to the destruction of most SWCNTs after ultrasonic treatment. These findings suggest that the structure of SWCNT films without cold pressing treatment is loose, insufficiently dense, and lacks stability.

In contrast, Fig. 6g–i showcases TEM images of films after cold pressing treatment. In Fig. 6g, even after ultrasonic treatment, the surface morphology of the film reveals a considerably compact structure with a remarkably smooth surface. Only a few interfaces with evident orientation can be characterized, indicating both high density and strong anisotropy of the film. Figure 6h zooms in on one of the interfaces, confirming SWCNTs by their diameters and displaying lattice information under a specific TEM observation angle. Figure 6i magnifies another interface, representing a local area with the random orientation of a few SWCNTs, which is a normal occurrence after cold pressing, and validates the characteristics of high density and orientation. This evidence confirms that the film after cold pressing exhibits

a dense structure, strong anisotropy, and considerable stability, which are fundamental reasons for its superior thermoelectric performance and practical utility. We further conducted Hall measurements on SWCNT films before and after cold pressing, as shown in Supplementary Fig. 7b. After cold pressing, the significant increase in film conductivity mainly stems from the simultaneous increase in $n_p$ and $\mu$ compared to the pristine film. The increase in $n_p$ and $\mu$ is primarily due to the tight connections between SWCNTs after cold pressing, which increases the contact area between SWCNTs. The reduction in interfaces decreases carrier scattering, making carriers in the film more transportable. Additionally, the compression of carriers within a certain space inevitably leads to an increase in $n_p$, as mentioned earlier[63–65].

## Flexibility, stability, and device performance

We also conducted bending tests on the film to evaluate the stability and flexibility during the test, and fabricated a six-legged device using the as-fabricated films, as shown in Fig. 7. Figure 7a compares the measured normalized resistance $R/R_0$ as a function of bending time for pristine, cold-pressed, and rolled $0.1\,\mathrm{M\,L^{-1}}$ NaBH$_4$-treated SWCNT films, and Fig. 7b shows a bent SWCNT film on a circular tube with a radius of 7 mm. Within 2000 bends, the maximum resistance error for the NaBH$_4$-treated film is 8.5 %. The resistance errors for the cold-

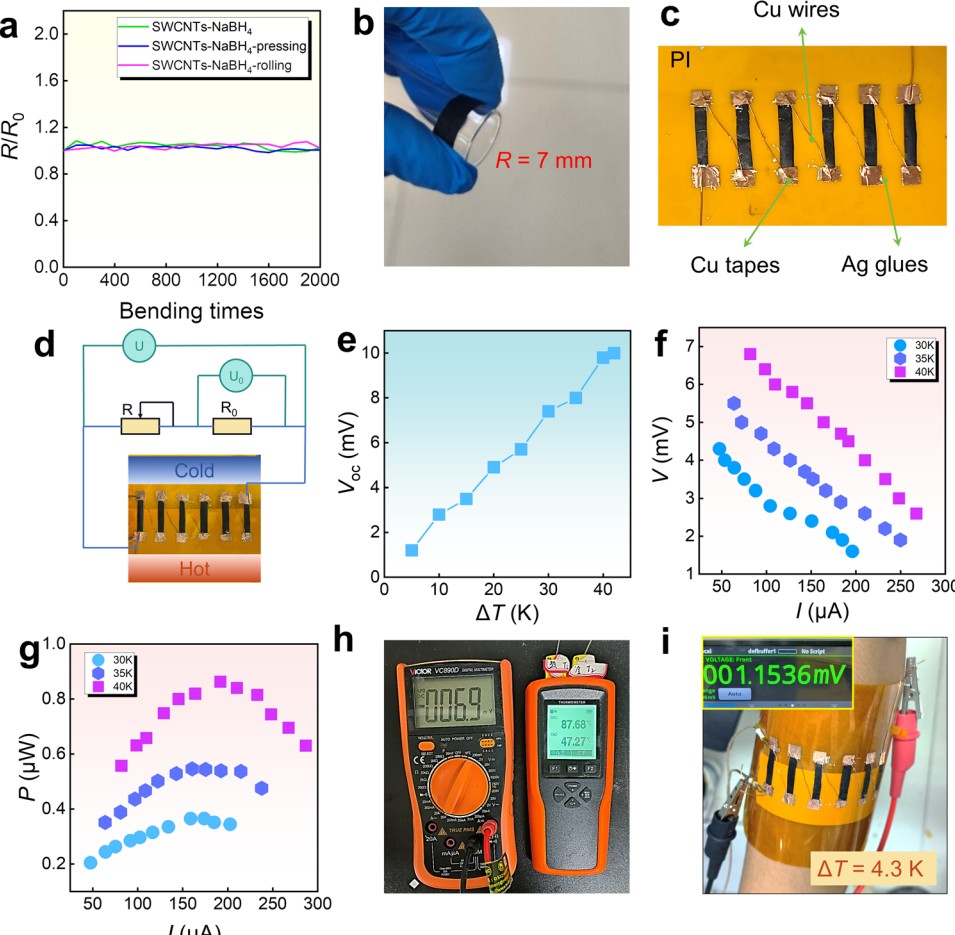

**Fig. 7 | Flexibility of SWCNT films and their devices with measured output performance. a** Normalized resistance $R/R_0$ as a function of bending time for pristine, cold-pressed, and rolled $0.1\,\mathrm{M\,L^{-1}}$ NaBH$_4$-treated SWCNT films. **b** Photograph illustrating a bent SWCNT film. **c** Photo of a six-legged device fabricated using NaBH$_4$-treated and cold-pressed SWCNT films. Here PI is abbreviated from polyimide. **d** Photograph illustrating the connection of the SWCNT film device

for cold side, hot side, and circuit. **e** Open-circuit voltage $V_{oc}$ of the six-legged device as a function of temperature difference $\Delta T$. **f** Output voltage $V$ and **g** output power $P$ of the device as a function of the current $I$ under $\Delta T$s of 30, 35, and 40 K. **h** Photograph illustrating the $V$ of the device after adjusting resistance. **i** Photograph illustrating the wearability of the device and the generated voltage while wearing the device.

pressed and rolled films are within 5% and 8%, respectively. To further verify the outstanding thermoelectric performance of the as-fabricated SWCNT film, we used the $NaBH_4$-SWCNTs-cold-pressed films to assemble a six-legged F-TED, as shown in Fig. 7c. Thermal and cold sources were added to the ends of the film to form the hot and cold sides, and the F-TED was connected to a voltage/resistance meter, as shown in Fig. 7d. To obtain an accurate $\Delta T$, a copper strip is used on the hot side for efficient heat conduction, ensuring uniform heat flow at both sides. On the cold side, an insulating layer is employed to isolate the impact of thermal radiation. As shown in Fig. 7e, the $V_{oc}$ generated by the F-TED increases with increasing $\Delta T$. At a $\Delta T$ of 40 K, the maximum $V_{oc}$ is 9.8 mV. Subsequently, a load resistor was connected to the F-TED to form a complete circuit and measured the $P$ of the SWCNT film device in the temperature range of 300–373 K. Figure 7f illustrates the inverse relationship between $V$ and $I$ for the device at $\Delta T$s of 30, 35, and 40 K. Figure 7g shows the functional relationship between $P$ and $I$ for $\Delta T$s of 30, 35, and 40 K. When $I$ is approximately 192 μA and $\Delta T$ is 40 K, a maximum $P$ of 0.86 μW can be achieved. Figure 7h represents the experimental data with an output voltage of approximately 6.9 mV at a $\Delta T$ of 40 K. When the film device was worn on the forearm, as shown in Fig. 7i. A $\Delta T$ of 4.3 K between the human skin and the environment can result in a voltage of 1.15 mV. The maximum power density reaches 2996 μW cm$^{-2}$ at a $\Delta T$ of 40 K, which is calculated by dividing the output power by the number of legs and the cross-sectional area of the F-TED. The as-achieved power density is highly competitive compared with reported values (Supplementary Table 4), demonstrating the applicability of the F-TED as a temperature sensor.

To verify the air stability of SWCNT films and their devices, we conducted repeated tests on the performance of SWCNT films after 3 months of exposure to the air, as shown in Supplementary Fig. 15a. After being placed in the air for 3 months, the $\sigma$ of the film was slightly increased. However, the $S$ decreased, possibly due to the increased defects caused by SWCNTs absorbing oxygen in the air, ultimately leading to a decrease in thermoelectric performance ($S^2\sigma$ of 13 μW cm$^{-1}$ K$^{-2}$). However, it remained much higher than that of the untreated SWCNT film ($S^2\sigma$ of 1.6 μW cm$^{-1}$ K$^{-2}$). In fact, any flexible thermoelectric material or device exposed to air for an extended period will experience a decline in performance[5], which is understandable. In addition, it should be noted that the treatment process we employed here (exposure to air for 3 months) is relatively extreme and aimed at further exploring the stability of the film, therefore the decrease of $S^2\sigma$ is acceptable. For the stability of the flexible device (Supplementary Fig. 15b), after a period of time, the stability of the six-legged device also deteriorates as the performance of the SWCNT film declines. With a $\Delta T$ at 40 K, $V_{oc}$ decreases to 6.2 mV, but the trend of voltage variation with temperature difference remains the same as before, primarily due to the decrease in the thermoelectric performance of the SWCNT film. Therefore, regular maintenance and care are necessary for the daily use of F-TEDs, and considering subsequent encapsulation can effectively enhance film and device stability[50].

In this study, the thermoelectric performance of flexible SWCNT films was improved by a simple "triple treatment" approach. Optimizing ultrasonic dispersion ensures uniform assembly of SWCNT films, and improves the film conductivity. Subsequent $NaBH_4$ treatment reduces surface defects and enhances the $S$. Cold pressing further densifies the film, maintaining a high $S$ and achieving a record power factor of 20.29 μW cm$^{-1}$ K$^{-2}$. The resulting SWCNT films exhibit structural stability, no rebound, and remarkable flexibility. A six-legged F-TED based on the as-fabricated films demonstrates a maximum output power of 0.86 μW and a maximum power density of 2996 μW cm$^{-2}$ at a $\Delta T$ of 40 K. This approach advances SWCNT films as high-performance thermoelectric materials for wearable devices and this "triple treatment" method could be used to prepare n-type SWCNT films.

## Methods

### Materials
SWCNTs (purity >95%, TNSR type, diameter: 1–2 nm) were purchased from Chengdu Zhongke Times Nano Energy Tech Co., Ltd. $NaBH_4$ was purchased from Shanghai Lingfeng Chemical Reagent Co., Ltd. The filter membrane, with a pore size of 0.45 μm, was obtained from Tianjin Jinteng Experimental Equipment Co., Ltd. Anhydrous ethanol and acetone were purchased from Shanghai Lingfeng Chemical Reagent Co., Ltd and Shanghai Shenbo Chemical Co., Ltd, respectively.

### Film fabrication
Initially, SWCNTs were introduced into ethanol and stirred at 600 rpm for 1 h to create a well-dispersed solution with a fixed concentration of 1 mg mL$^{-1}$. Subsequently, ultrasonication was carried out for 5, 10, 15, 20, 25, 30, 35, and 40 min in an ice bath to yield SWCNT dispersion solutions with varying ultrasonic times. A specific volume of the SWCNT dispersion solution was then taken and dried in a custom polytetrafluoroethylene mold at 60 °C for 18 h, resulting in SWCNT films with different ultrasonic times. For substrate preparation, a 2 × 3 cm glass substrate underwent sequential ultrasonic cleaning with deionized water, acetone, and ethanol. It was further treated with an ultraviolet ozone cleaner for 30 min. Following this, the SWCNT films were immersed in $NaBH_4$ solutions of varying concentrations for 30 min at room temperature. The films were then rinsed in deionized water three times to eliminate residual solvent and subsequently vacuum-dried for 15 h at 60 °C, producing the $NaBH_4$-treated SWCNT films (SWCNTs-$NaBH_4$). $NaBH_4$ concentrations were set at 0, 0.05, 0.1, 0.15, and 0.2 mol L$^{-1}$ (hereinafter denoted as M L$^{-1}$). Finally, the $NaBH_4$-treated SWCNT films underwent pressing under a vertical pressure of 3 MPa for 10 min, resulting in cold-pressed SWCNT films (SWCNTs-$NaBH_4$-pressing). Alternatively, the treated SWCNT films (SWCNTs-$NaBH_4$) were subjected to repeat rolling on a rolling machine ten times, leading to rolled SWCNT films (SWCNTs-$NaBH_4$-rolling). The untreated SWCNT films followed the same procedures as described above.

### Flexibility testing and device fabrication
One end of the pristine, cold-pressed, and rolled SWCNT films treated with 0.1 M L$^{-1}$ $NaBH_4$ were affixed to a circular tube with a 7 mm radius. The opposite end of the film was pressed with a finger to conform to the circular tube surface and then released to complete one bending cycle. This bending procedure was multiply repeated. A four-legged device is constructed with four pieces of 23 mm × 4 mm SWCNTs-0.1 M L$^{-1}$ $NaBH_4$-pressing films, with an average thickness of around 1.2 μm. The films are securely attached to a polyimide substrate using a silver paste, and interconnections between the films are established using 0.3-mm-diameter copper wires.

### Thermoelectric performance testing
The in-plane $\sigma$ and $S$ of the SWCNT films were measured at room temperature using the Netsch SBA 458 system. The in-plane thermal diffusivity $D$ was measured at room temperature using LaserPIT from ADVANCE RIKO, Inc. The specific heat capacity $C_p$ adopts the theoretical values of SWCNTs. The surface and cross-sectional morphologies of the films were studied through scanning electron microscopy (SEM S4800). The elemental distributions in SWCNT films were characterized by X-ray energy-dispersive spectroscopy (EDS, OXFORD Xplore). Raman spectroscopy analysis, spanning from 100 to 3000 cm$^{-1}$, was performed using a confocal Raman microscope (LabRAM HR Evolution) equipped with a 553 nm laser. The XPS analysis was conducted by multipurpose X-ray photo-emission spectroscopy (Thermo ESCALAB 250, country). The Hall coefficient $R$ data were measured using the Van der Pauw method (CH-70, CH-magnetoelectricity Technology Co., Ltd., China) under a magnetic field up to 100 mT. The $n_p$ and $\mu$ were determined by $n_p = 1/eR$ and $\mu = \sigma R$, respectively. The nanostructural features of SWCNTs were observed by probe-corrected scanning

transmission electron microscopy (STEM, Hitachi HF5000). The ultrasonic treatment before TEM characterizations was based on an ultrasonic machine (FXP14+ Model, Unisonics Australia Pty Ltd) with durations of 30 min. The transducer frequency is 40 kHz, and the ultrasonic power is 200 W. The $V$ of the devices was recorded using a multimeter, and the $\Delta T$ across the devices was measured using a thermocouple.

## Calculation

First-principle calculations were performed based on DFT with all electron-projected augmented wave methods, as implemented in the Vienna Ab initio Simulation Package (VASP)[66–68]. Semi-local generalized gradient approximation (GGA) with the fully relativistic Perdew–Burke–Ernzerhof (PBE) exchange-correlation functional was employed[69]. The Brillouin zone was sampled by a Monkhorst–Pack **k**-mesh spanning less than 0.06/Å$^3$ for structural relaxation, and a denser **k**-mesh spanning less than 0.03/Å$^3$ for calculating static self-consistency and density-of-state (DOS). The wave functions were expanded in a plan-wave basis with a cut-off energy of 500 eV. All atoms were allowed to relax in their geometric optimizations until the Hellmann–Feynman force was less than $1 \times 10^{-3}$ eV Å$^{-1}$, and the convergence criterion for the electronic self-consistent loop was set to $1 \times 10^{-7}$ eV. The electron band structures were calculated along the line-mode **k**-path based on Brillouin path features indicated by the AFLOW framework[70].

## Data availability

The data generated in this study are provided in the Supplementary Information/Source Data file. Source data are provided with this paper.

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

## Acknowledgements

This work is financially supported by the National Natural Science Foundation of China (No. 52272040), the State Key Laboratory of Materials-Oriented Chemical Engineering (SKL-MCE-23A04), and the Jiangsu Specially-Appointed Professor Program (Q. Liu). This work also received financial support from the Australian Research Council, HBIS-UQ Innovation Centre for Sustainable Steel project, and the QUT Capacity Building Professor Program (Z.-G. Chen). Z.-G. Chen and M. Li acknowledge the National Computational Merit Allocation Scheme 2024 (wk98), supported by the National Computational Infrastructure, for providing computational resources and services. This work was

enabled by the use of the Central Analytical Research Facility hosted by the Institute for Future Environments at QUT.

## Author contributions

Y.-M. Liu & X.-L. Shi contributed equally to this work. Z.-G. Chen and Q. Liu supervised the project and conceived the idea. Y.-M. Liu and X.-L. Shi designed the experiments and wrote the manuscript. Y.-M. Liu, T. Wu, H. Wu, and D.-Z. Wang performed the sample synthesis, structural characterization, and thermoelectric transport property measurements. M. Li and X.-L. Shi undertook the theoretical work. Y. Mao conducted the TEM measurements and T. Cao conducted the in-plane $\kappa$ of the SWCNT films. X.-L. Shi, M. Li, W.-D. Liu, Z.-G. Chen, and Q. Liu undertook the thermoelectric performance evaluation. All the authors discussed the results and commented on the manuscript. All authors have given approval to the final version of the manuscript.

## Competing interests

The authors declare no competing interests.
