## [Peer Review File · Nature Communications]

Boosting thermoelectric performance of single-walled carbon nanotubes based films through rational triple treatmentsREVIEWER COMMENTS

Reviewer #1 (Remarks to the Author):

Some questions and comments listed below need further consideration by authors to improve the quality of the manuscript.

1. SWCNTs are noted for high thermal conductivity which is detrimental for their use in getting high heat-to-electricity conversion efficiency. High as its power factor of the triple-treated SWCNT film is, the thermal conductivity of resulting SWCNT film still should be measured although precise data are difficult to obtain, particularly that along the in-plane direction.
2. DFT calculation is recommended to theoretically study the significant enhancement in thermoelectric properties of SWCNTs after triple treatments.
3. Although XPS analysis finds no sodium borohydride left in the SWCNTs, further evidence like from EDS mapping is necessary to validate it.
4. What is the optimal treatment time of the cold-pressing process?
5. As for the writing, authors need to pay attention to the following expressions (not limited to these examples).
Format of the author list in references is inconsistent, such as Xu, S. et al. in ref. 8, Blackburn, J.L., Ferguson, A.J., Cho, C. & Grunlan, J.C. in ref. 9, and Tserengombo, B., Jeong, H., Dolgor, E., Delgado, A. & Kim, S. in ref. 13.
... deviceization ...
Variables (Fig. 5) should be expressed by italic letters.

Reviewer #2 (Remarks to the Author):

This manuscript report record-high thermoelectric performance of flexible SWCNT films using “triple treatments” without any doping or hybridization. The method is simple and effective. However, the characterizations are insufficient. Some major revisions need to be done prior to consideration of publishing.

1. The electrical conductivity of pristine SWCNT film is prominently high (~ 3000 S/cm), which is higher than most reported values, even higher than most of the conductivities of optimized SWCNT films as reported in Table S1. Is there any special treatment for this material?
2. “The σ of the SWCNTs-NaBH₄ film increases to 14300 S cm⁻¹ after cold-pressing, nearly 12 times the original value, while it increases to 11000 S cm⁻¹ after rolling, about 9 times the original value.” Why does the conductivity increase so much? Is it mainly due to the increase in carrier concentration or the increase in carrier mobility? Further analysis is needed, such as Hall measurements, for example.
3. Line 270 ~ 274, the authors mentioned that cold-pressing process leads to an increase in defects and a tighter connection between SWCNT layers. These two aspects have opposite effects on the σ . The significant increase in conductivity caused by the cold-pressing process requires further analysis.
4. In Figure 3d, Why does the 2D peak first decrease and then increase with the increase in NaBH₄ concentration?
5. Line 317, the authors mentioned that the XPS peaks for C1s and O1s are 280.8 eV and 528 eV, respectively. These peaks appear to have not undergone charge

correction. Normally, the XPS peaks for C1s and O1s are 284.8 eV and 532 eV, respectively. Therefore, the shift of 3.3 eV in Figure 5b needs further consideration.

6. How were the TEM samples prepared? If these samples were prepared by drop-casting solutions onto copper grids, it seems to me that using this method to demonstrate the orientation of the film is not very convincing.

7. Can this "triple treatments" method be used to prepare n-type SWCNT films?

8. How about the air stability of the σ and S for the SWCNT films? Also, How about the air stability of the output power of the device?

Reviewer #3 (Remarks to the Author):

The authors boosted thermoelectric performance and stability of SWCNT-based flexible films and devices through rational triple treatments. The manuscript is well organized, and the triple treatments used is effective to enhance TE properties of SWCNT-based films. I think this manuscript can be published after major revision. The comments are as follows:

1. It seems that the longer the ultrasonication time, the smaller value of the PF for the SWCNT-based films.

2. Hall effect of the SWCNT-based films treated with different content of NaBH₄ solution is suggested to be added.

3. Both electrical conductivity and Seebeck coefficient have been improved simultaneously when the SWCNT-based films treated with high content of NaBH₄ solution (0.15 M-0.2 M). However, I wonder that how about the consequence of the NaBH₄ solution with content higher than that of 0.2M.

4. Whether the prepared SWCNT-based flexible films has the anisotropy after rolling treatment.

5. The style of references should be uniformed.

Response to reviewers

Reviewer #1:

General comments: Some questions and comments listed below need further consideration by authors to improve the quality of the manuscript.

Author reply: We highly value your constructive and supportive feedback on our work.

Comment 1. *SWCNTs are noted for high thermal conductivity which is detrimental for their use in getting high heat-to-electricity conversion efficiency. High as its power factor of the triple-treated SWCNT film is, the thermal conductivity of resulting SWCNT film still should be measured although precise data are difficult to obtain, particularly that along the in-plane direction.*

Author reply: Following the suggestion, we have measured the thermal conductivity (κ) and ZT values of SWCNTs films before and after triple-treatments, as illustrated in **Figure R1** (**Figure S8** in the revised Supporting Information on **Page 13**). We have added more related discussions as “However, after triple-treatments, there is a slight increase in the κ of SWCNT films, but the final ZT value remains significantly higher than that of the pristine film (specific values provided in **Figure S8**). The increase in κ of SWCNT films after pressing is attributed to the densification of the initially porous film structure, facilitating heat transfer. However, as the electrical transport properties become superior after pressing, offsetting the adverse effects of increased κ , the final ZT value is enhanced” on **Page 12** of the revised manuscript.

Figure R1 (Figure S8 in the revised Supporting Information). Thermal conductivity κ and ZT of untreated SWCNT film and 0.1 M L^{-1} NaBH₄-treated SWCNT film after cold-pressing.

Comment 2. *DFT calculation is recommended to theoretically study the significant enhancement in thermoelectric properties of SWCNTs after triple treatments.*

Author reply: Following the suggestion, first-principles density-functional theory (DFT) calculations were conducted for SWCNTs with various chiral structures, both with and without defects, as depicted in **Figure R2** (Figure 4 in the revised manuscript on Page 35). We have added more related discussions as “To theoretically demonstrate the variation in the thermoelectric properties of SWCNT films, we conducted first-principles density functional theory (DFT) calculations on SWCNTs with different chiral structures. Generally, SWCNTs exhibit three chiral structures ¹. **Figures 4a-c** illustrate the

calculated band structures of pristine SWCNTs with chiral indices $(2n+m) \bmod 3 = 0, 1, \text{ and } 2$, respectively. From the calculations, it is evident that SWCNTs with chiral index $(2n+m) \bmod 3 = 0$ overlap in the conduction and valence bands, exhibiting metallic properties with almost no bandgap, thus demonstrating electron conduction characteristics (n-type). On the other hand, SWCNTs with chiral indices $(2n+m) \bmod 3 = 1$ and 2 exhibit indirect bandgaps >0.6 , indicating semiconductor characteristics. Since the SWCNTs used in the experiment are p-type as indicated by the measured positive S , our SWCNTs exhibit semiconductor behaviors, specifically $(2n+m) \bmod 3 = 1$ or 2 . **Figures 4d-f** show the calculated band structures of SWCNTs with a single carbon vacancy defect for chiral indices $(2n+m) \bmod 3 = 0, 1, \text{ and } 2$, respectively. It can be observed that upon removing one carbon atom to form a vacancy defect, the bandgaps of SWCNTs with $(2n+m) \bmod 3 = 1$ or 2 decrease, enhancing carrier transition capabilities and in turn leading to an increase in σ and a decrease in the S . Considering that after treatment with a certain concentration of NaBH_4 in the experiment, the defects in SWCNT films decrease, resulting in an increase in the bandgap and S , which is consistent with experimental results. We also calculated the charge distribution functions and electron localization functions before and after the formation of a vacancy defect, as shown in **Figures 4g-j**. For p-type SWCNTs, with an increase in defect content, the carrier density around the defect increases. Therefore, after treatment with $0.1 \text{ M L}^{-1} \text{ NaBH}_4$, the defects in SWCNT films decrease, leading to a decrease in n_p and σ' on **Page 11** of the revised manuscript.

In addition to this, we have added more discussions as “Simultaneously, theoretical calculations also indicate that the introduction of defects on SWCNTs induces a reduction in the bandgap of SWCNT films, increased in n_p , and in turn a decrease in S and an increase in σ , as shown in **Figure 4**. Therefore, the potential generation of defects during the cold-pressing process is also a reason for the

change in the thermoelectric properties of SWCNT films” on **Page 13** of the revised manuscript.

Figure R2 (Figure 4 in the revised manuscript). Calculation results of SWCNT films at different (n, m) indices with and without defects. Calculated band structures of pristine SWCNTs for (a) type $(2n+m) \bmod 3 = 0$, (b) type $(2n+m) \bmod 3 = 1$, and (c) type $(2n+m) \bmod 3 = 2$; and calculated band structures of SWCNTs with a carbon vacancy for (d) type $(2n+m) \bmod 3 = 0$, (e) type $(2n+m) \bmod 3 = 1$, and (f) type $(2n+m) \bmod 3 = 2$. (g) Charge density distribution diagram and (h) electron

localization function diagram of SWCNTs for type $(2n+m) \bmod 3 = 1$ with and without carbon vacancy; and (i) charge density distribution diagram and (j) electron localization function diagram of SWCNTs for type $(2n+m) \bmod 3 = 2$ with and without carbon vacancy. The positions with carbon vacancies are indicated with arrows.

We have also added the calculation details to the experimental part on **Page 21** of the revised manuscript as “First-principle calculations were performed based on density-functional theory (DFT) with all electron projected augmented wave (PAW) method, as implemented in the Vienna Ab initio Simulation Package (VASP) ²⁻⁷. Semi-local generalized gradient approximation (GGA) with the fully relativistic Perdew-Burke-Ernzerhof (PBE) exchange correlation functional was employed ⁸. The Brillouin zone was sampled by a Monkhorst-Pack **k**-mesh spanning less than $0.06/\text{\AA}^3$ for structural relaxation, and a denser **k**-mesh spanning less than $0.03/\text{\AA}^3$ for calculating static self-consistency and density-of-state (DOS). The wave functions were expanded in a plan-wave basis with a cut-off energy of 500 eV. All atoms were allowed to relax in their geometric optimizations until the Hellmann–Feynman force was less than $1 \times 10^{-3} \text{ eV} \cdot \text{\AA}^{-1}$, and the convergence criterion for the electronic self-consistent loop was set to $1 \times 10^{-7} \text{ eV}$. The electron band structures were calculated along the line-mode **k**-path based on Brillouin path features indicated by the AFLOW framework ⁹”.

Comment 3. *Although XPS analysis finds no sodium borohydride left in the SWCNTs, further evidence like from EDS mapping is necessary to validate it.*

Author reply: To assess the presence of residual sodium borohydride in the SWCNTs, we conducted

energy-dispersive spectrometry (EDS) characterization, depicted in **Figure R3** (**Figure S14** in the revised Supporting Information on **Page 19**). The EDS diagram reveals carbon (C) as the primary element in SWCNT films before and after treatment. Considering the error range, it is reasonable to conclude that there was essentially no NaBH₄ residue. Also, the differences in signal contrast observed on the EDS maps mainly stem from surface morphology variations of the film samples. We have added more discussions about this as “Following NaBH₄ treatment, no additional elements such as B or Na are introduced, suggesting that NaBH₄ solvent treatment does not adsorb onto the surface of SWCNTs but rather induces valence state and defect regulation, as shown in **Figure S14**” on **Page 15** of the revised manuscript.

Figure R3 (**Figure S14** in the revised Supporting Information). Energy-dispersive spectroscopy (EDS) maps of (a) C, (b) Na, (c) O elements of untreated SWCNT films, and (d) C, (e) Na, (f) O of 0.2 M L⁻¹ NaBH₄-treated SWCNT films.

Comment 4. *What is the optimal treatment time of the cold-pressing process?*

Author reply: Due to the gradual pressure build-up during the cold pressing procedure, the duration of the process may vary. Optimal results are achieved by maintaining the pressure for 10 minutes once the desired pressure reaches, leading to a significant enhancement in thermoelectric performance. When the pressing duration is too short, the film has not reached a dense state, thus limiting the enhancement of its thermoelectric performance. Also, prolonged pressure will result in strong adhesion issues between the film and the pressed substrate, impeding the formation of a complete and high-quality SWCNT film.

Comment 5. *As for the writing, authors need to pay attention to the following expressions (not limited to these examples). Format of the author list in references is inconsistent, such as Xu, S. et al. in ref. 8, Blackburn, J.L., Ferguson, A.J., Cho, C. & Grunlan, J.C. in ref. 9, and Tserengombo, B., Jeong, H., Dolgor, E., Delgado, A. & Kim, S. in ref. 13. ... deviceization ... Variables (Fig. 5) should be expressed by italic letters.*

Author reply: In terms of writing, we meticulously checked for spelling errors, ensuring consistency in the formatting of author lists within references, as well as representing all variables using italicized letters. Revised references are shown on **Pages 23-24** of the revised manuscript.

“8. Xu, S. et al. Conducting polymer-based flexible thermoelectric materials and devices: from mechanisms to applications. *Prog. Mater. Sci.* **121**, 100840 (2021).

9. Blackburn, J.L., Ferguson, A.J., Cho, C. & Grunlan, J.C. Carbon-nanotube-based thermoelectric materials and devices. *Adv. Mater.* **30**, 1704386 (2018).

13. Tserengombo, B., Jeong, H., Dolgor, E., Delgado, A. & Kim, S. Effects of functionalization in different conditions and ball milling on the dispersion and thermal and electrical conductivity of MWCNTs in aqueous solution. *Nanomaterials* 11, 1323 (2021).”

We also revised the corresponding description as “Flexibility of SWCNT films and their devices with measured output performance” on **Page 40** of the revised manuscript.

Reviewer #2:

General comments: This manuscript reports record-high thermoelectric performance of flexible SWCNT films using “triple treatments” without any doping or hybridization. The method is simple and effective. However, the characterizations are insufficient. Some major revisions need to be done prior to consideration of publishing.

Author reply: We appreciate the constructive and insightful comments.

Comment 1. *The electrical conductivity of pristine SWCNT film is prominently high (~ 3000 S/cm), which is higher than most reported values, even higher than most of the conductivities of optimized SWCNT films as reported in Table S1. Is there any special treatment for this material?*

Author reply: In the absence of additional processing, the pristine SWCNTs exhibit high electrical conductivity, likely attributing to their high purity but with certain inherent defects, as illustrated in **Figure R4 (Figure 3d** in the revised manuscript on **Page 34**). The pristine SWCNT films display a high G/D peak intensity ratio of ~ 40, indicating high purity and few defects. Higher purity ensures structural stability of the SWCNTs with increased charge carrier mobility, while a certain concentration of initial defects, such as common carbon vacancies, effectively enhance the overall carrier concentration of the material, thus achieving higher initial conductivity. To elaborate on the relationship between defects and carrier concentration, we employed the first-principles calculations, as detailed on **Pages 11 & 14** of the revised manuscript. The calculations results are also shown in **Figure R2 (Figure 4** in the revised manuscript on **Page 35**).

Figure R4 (Figure 3d in the revised manuscript). Raman spectra of SWCNT films treated with different concentrations of NaBH₄.

Comment 2. “The σ of the SWCNTs-NaBH₄ film increases to 14300 S cm⁻¹ after cold-pressing, nearly 12 times the original value, while it increases to 11000 S cm⁻¹ after rolling, about 9 times the original value.” Why does the conductivity increase so much? Is it mainly due to the increase in carrier concentration or the increase in carrier mobility? Further analysis is needed, such as Hall measurements, for example.

Author reply: Following the suggestions, we conducted Hall measurements on SWCNT films before and after cold pressing to analyze the significant increase in film conductivity, as depicted in **Figure R5** (Figure S7b in the revised Supporting Information on Page 12). We have added more discussions as “However, after cold pressing, the connections between SWCNTs in the film become tighter, which

significantly improves the σ of the film. From the intensity of the 2D peak, it can be observed that while maintaining the SWCNT structure, both the 2D peak ratio and the thickness of the film significantly decrease. After the film thickness decreases significantly, the overall volume of the film decreases substantially. However, since the change in the total carrier quantity is not significant (due to a relatively constant defect concentration), the sharp increase in n_p is primarily due to the significant decrease in the overall volume of the film (**Figure S7b**). Therefore, the densification of SWCNT films after cold pressing plays a dominant role in the high conductivity of SWCNT films” on **Page 14** of the revised manuscript, and “We further conducted Hall measurements on SWCNT films before and after cold pressing, as shown in **Figure S7b**. After cold pressing, the significant increase in film conductivity mainly stems from the simultaneous increase in n_p and μ compared to the pristine film. The increase in n_p and μ is primarily due to the tight connections between SWCNTs after cold pressing, which increases the contact area between SWCNTs. The reduction in interfaces decreases carrier scattering, making carriers in the film more transportable. Additionally, the compression of carriers within a certain space inevitably leads to an increase in n_p , as mentioned earlier¹⁰⁻¹²” on **Page 16** of the revised manuscript.

Figure R5 (Figure S7b in the revised Supporting Information). Room-temperature n and μ during triple treatments.

Comment 3. Line 270 ~ 274, the authors mentioned that cold-pressing process leads to an increase in defects and a tighter connection between SWCNT layers. These two aspects have opposite effects on the σ . The significant increase in conductivity caused by the cold-pressing process requires further analysis.

Author reply: The cold pressing process induces increase in defects and tighter interconnections between SWCNT layers. Both aspects positively contribute to the enhancement of conductivity. After cold pressing, the significant increase in film conductivity mainly stems from the simultaneous increase in n_p and μ compared to the pristine film. The increase in n_p and μ is primarily due to the tight connections between SWCNTs after cold pressing, which increases the contact area between SWCNTs. The reduction in interfaces decreases carrier scattering, making carriers in the film more transportable.

Additionally, the compression of carriers within a certain space inevitably leads to an increase in n_p . This is because after the film thickness significantly decreases, the overall volume of the film substantially decreases. However, since the change in the total carrier quantity is not significant (due to a relatively constant defect concentration), the sharp increase in n_p is primarily due to the significant decrease in the overall volume of the film. In terms of the relationship between the defects and the conductivity, the G/D peak ratio of SWCNT films decreases slightly after cold pressing (**Figure 5**), indicating a slight increase in defects that also contributes to an enhancement in the σ of the film. The theoretical calculations also indicate that the introduction of defects on SWCNTs induces a reduction in the bandgap of SWCNT films, increased in n_p , and in turn a decrease in S and an increase in σ , as shown in **Figure 4**. We have added these discussions on **Pages 14 & 16** of the revised manuscript.

Comment 4. *In Figure 3d, why does the 2D peak first decrease and then increase with the increase in NaBH₄ concentration?*

Author reply: We have revised the corresponding discussions as “With increasing the concentration of NaBH₄ treatment, the intensity of the 2D peak was generally increased first and then decreased. This is mainly attributed to the varying effects of interlayer separation among films treated with different concentrations of NaBH₄” On **Page 9** of the revised manuscript.

Comment 5. *Line 317, the authors mentioned that the XPS peaks for C1s and O1s are 280.8 eV and 528 eV, respectively. These peaks appear to have not undergone charge correction. Normally, the XPS*

peaks for C1s and O1s are 284.8 eV and 532 eV, respectively. Therefore, the shift of 3.3 eV in Figure 5b needs further consideration.

Author reply: Following the suggestion, we have modified the XPS diagram, as shown in **Figure R6** (**Figures 6a-b** in the revised manuscript on **Page 37**). After charge correction, the position of the C1s peak is at 284.8 eV and the position of the O1s peak is at 532.2 eV, as show in **Table R1** (**Table S3** in the revised Supporting Information on **Page 4**). We have added corresponding description as “Charge correction for C1s and O1s peaks was performed against untreated SWCNTs” on **Page 15** of the revised manuscript.

Figure R6 (**Figure 6a-b** in the revised manuscript). (a) X-ray photoelectron spectroscopy (XPS) patterns with a broad range, and (b) Corresponding magnified C1s patterns of pristine, cold-pressed, and rolled SWCNT films with and without 0.1 M L⁻¹ NaBH₄ treatment.

Table R1 (**Table S3** in the revised Supporting Information). X-ray photoelectron spectroscopy (XPS) characterized the position of atomic in different SWCNTs films. Here “SWCNTs-pressing” indicates

the SWCNTs films after cold pressing, “SWCNTs-rolling” indicates the SWCNTs films after rolling, and “SWCNTs-0.1” indicates the SWCNTs films with 0.1 M L⁻¹ NaBH₄ treatment.

Binding energy(eV)	SWCNTs	SWCNTs-pressing	SWCNTs-rolling	SWCNTs-0.1	SWCNTs-0.1-pressing	SWCNTs-0.1-rolling
C1s	284.8	284.9	284.8	288.1	284.7	284.9
O1s	532.2	532.4	532.0	535.6	532.2	532.3

Comment 6. *How were the TEM samples prepared? If these samples were prepared by drop-casting solutions onto copper grids, it seems to me that using this method to demonstrate the orientation of the film is not very convincing.*

Author reply: The TEM samples were prepared by injecting SWCNT film onto copper grids after 30 minutes of regular sonication in ethanol. We used a cell disruptor ultrasonicator for sonication in our experiments, which delivers much higher ultrasonic power compared to standard sonication. We believe standard ultrasonic waves do not disrupt the aggregation pattern of the film.

Comment 7. *Can this "triple treatments" method be used to prepare n-type SWCNT films?*

Author reply: This is a very valuable suggestion. We believe that the “triple treatment” method could also be applicable for preparing n-type SWCNT films. This is because the primary impact of the processing technique we designed on the film is its ability to significantly reduce the overall volume of the film, thereby substantially increasing its density and making the film internally dense. This, in turn, greatly enhances conductivity. Particularly, the Seebeck coefficient of thermoelectric materials

generally remains largely unaffected by material structure¹³. While cold pressing increases the defect density in the film to a limited extent, the significant improvement in film density is considered the primary reason for the enhancement of thermoelectric performance. Therefore, we anticipate that cold pressing treatment for n-type SWCNT films would yield similar results. So, we have added corresponding description as “and this "triple treatments" method could be used to prepare n-type SWCNT films.” On **Page 19** of the revised manuscript.

Comment 8. *How about the air stability of the σ and S for the SWCNT films? Also, how about the air stability of the output power of the device?*

Author reply: Following the suggestion, we added the repeating tests on the performance of SWCNTs films after triple treatments and open-circuit voltage V_{oc} of the six-legged device as a function of temperature difference ΔT after placed for three months, as shown in **Figure R7** (**Figure S15** in the revised Supporting Information on **Page 20**). We have added more discussions as “To verify the air stability of SWCNT films and their devices, we conducted repeated tests on the performance of SWCNT films after three months of exposure to the air, as shown in **Figure S15a**. After being placed in the air for three months, the σ of the film was slightly increased. However, the S decreased, possibly due to the increased defects caused by SWCNTs absorbing oxygen in the air, ultimately leading to a decrease in thermoelectric performance ($S^2\sigma$ of $13 \mu\text{W cm}^{-1} \text{K}^{-2}$). It is worth mentioning that it remained much higher than that of the untreated SWCNT film ($S^2\sigma$ of $1.6 \mu\text{W cm}^{-1} \text{K}^{-2}$). Any flexible thermoelectric material or device exposed to air for an extended period will experience a decline in performance¹³, which is understandable. In addition, it should be noted that the treatment process we employed here (exposure to air for three months) is relatively extreme and aimed at further exploring

the stability of the film, therefore the decrease of $S^2\sigma$ is acceptable. For the stability of the flexible device (**Figure S15b**), after a period, the stability of the six-legged device also deteriorates as the performance of the SWCNT film declines. With a ΔT at 40 K, V_{oc} decreases to 6.2 mV, but the trend of voltage variation with temperature difference remains the same as before, primarily due to the decrease in the thermoelectric performance of the SWCNT film. Therefore, regular maintenance and care are necessary for the daily use of flexible thermoelectric devices, and considering subsequent encapsulation can effectively enhance film and device stability ¹⁴, on **Page 18** of the revised manuscript.

Figure R7 (Figure S15 in the revised Supporting Information). The air stability of the SWCNT films after triple treatments and the six-legged device after 3 months. **(a)** σ , S , and $S^2\sigma$ of SWCNT films with triple-treatments and **(b)** open-circuit voltage V_{oc} of the device as a function of temperature difference ΔT before and after being exposed to the air for three months.

Reviewer #3:

General comments: *The authors boosted thermoelectric performance and stability of SWCNT-based flexible films and devices through rational triple treatments. The manuscript is well organized, and the triple treatments used is effective to enhance TE properties of SWCNT-based films. I think this manuscript can be published after major revision. The comments are as follows:*

Author reply: We appreciate the constructive and supportive comments on our work.

Comment 1. *It seems that the longer the ultrasonication time, the smaller value of the PF for the SWCNT-based films.*

Author reply: Following the suggestion, we have revised the discussions as “By applying ultrasonication, the concentration and arrangement of SWCNTs were altered, leading to changes in conductivity. With a certain duration of ultrasonication (10 minutes), the original aggregated state of SWCNTs reached a stable bundled state. However, with further increasing the duration of ultrasonication, the arrangement of SWCNTs shifted from ordered back to disordered. Therefore, after reaching optimal performance, prolonged exposure to ultrasonication will lead to a decrease in the thermoelectric properties of SWCNT films” on **Page 8** of the revised manuscript.

Comment 2. *Hall effect of the SWCNT-based films treated with different content of NaBH₄ solution is suggested to be added.*

Author reply: Following the suggestion, we carried out Hall measurements on SWCNT films before and after treating with different contents of NaBH₄ solution, as shown in the **Figure R8 (Figure S7a**

of the revised Supporting Information on **Page 12**). We have added more discussions as “In addition, we conducted Hall measurements on the film samples (**Figure S7a**). The results indicate that with increasing the NaBH₄ concentration, the hole carrier concentration n_p of the film initially decreased and then leveled off. The S increased as the n_p decreased, consistent with experimental data. Furthermore, the carrier mobility μ of the film increased with increasing the NaBH₄ concentration. This is attributed to the weakening of carrier scattering among carriers due to the decrease in n_p ” on **Page 10** of the revised manuscript.

Figure R8 (**Figure S7a** of the revised Supporting Information). Room-temperature n and μ of the SWCNT films treated with different concentrations of NaBH₄.

Comment 3. Both electrical conductivity and Seebeck coefficient have been improved simultaneously when the SWCNT-based films treated with high content of NaBH₄ solution (0.15 M-0.2 M). However,

I wonder that how about the consequence of the NaBH₄ solution with content higher than that of 0.2M.

Author reply: Following the suggestion, we added two groups of NaBH₄ solution treatments with concentrations higher than 0.2 M L⁻¹, namely 0.25 M L⁻¹ and 0.3 M L⁻¹ NaBH₄ solution, respectively as shown in **Figure R9** (**Figure S4** of the revised Supporting Information on **Page 9**). After treatment with NaBH₄ concentrations exceeding 0.2 M L⁻¹, the conductivity and Seebeck coefficient continued to exhibit opposite trends. The Seebeck coefficient increased to 36 μV K⁻¹, which was lower than that of SWCNT films treated with 0.1 M L⁻¹, while the conductivity was lower, and there was no significant improvement in the overall power factor. We have added more discussions as: “At higher concentrations, there was no significant improvement in the properties of SWCNT films, as depicted in **Figure S4**” on **Page 9** of the revised manuscript.

Figure R9 (**Figure S4** of the revised Supporting Information). Thermoelectric performance of SWCNT films by different concentrations of NaBH₄ treatments. Room-temperature (a) electrical conductivity σ , (b) Seebeck coefficient S , and (c) power factor $S^2\sigma$ of SWCNT films as a function of NaBH₄ concentration. Here M is abbreviated from “mol”.

Comment 4. Whether the prepared SWCNT-based flexible films has the anisotropy after rolling

treatment.

Author reply: Similar to direct cold pressing, SWCNT film samples processed through rolling should exhibit analogous anisotropy. This is because rolling applies pressure consistently in the direction perpendicular to the film's in-plane direction during processing, similar to the conditions of direct cold pressing. Consequently, it can be anticipated that SWCNT film samples processed through rolling would display similar anisotropic characteristics. Considering that the overall thermoelectric performance of the film processed through rolling is inferior to that of the directly cold-pressed film, we did not give extensive discussions its structural changes.

Comment 5. *The style of references should be uniformed.*

Author reply: We have carefully checked and revised the references to ensure that the styles of references are uniform.

Reference

1. Blackburn, J.L., Ferguson, A.J., Cho, C. & Grunlan, J.C. Carbon-nanotube-based thermoelectric materials and devices. *Adv. Mater.* **30**, 1704386 (2018).
2. Kresse, G.Hafner, J. Ab initio molecular-dynamics simulation of the liquid-metal-amorphous-semiconductor transition in germanium. *Phys. Rev. B* **49**, 14251-14269 (1994).
3. Kresse, G.Hafner, J. Ab initio molecular dynamics for liquid metals. *Phys. Rev. B* **47**, 558-561 (1993).
4. Kresse, G.Furthmüller, J. Efficiency of *ab-initio* total energy calculations for metals and semiconductors using a plane-wave basis set. *Comp. Mater. Sci.* **6**, 15-50 (1996).
5. Kresse, G.Hafner, J. Norm-conserving and ultrasoft pseudopotentials for first-row and transition elements. *J. Phys. Condens. Mat.* **6**, 8245-8257 (1994).
6. Kresse, G.Furthmüller, J. Efficient iterative schemes for *ab initio* total-energy calculations using a plane-wave basis set. *Phys. Rev. B* **54**, 11169-11186 (1996).
7. Kresse, G.Joubert, D. From ultrasoft pseudopotentials to the projector augmented-wave method. *Phys. Rev. B* **59**, 1758-1775 (1999).
8. Perdew, J.P., Burke, K. & Ernzerhof, M. Generalized gradient approximation made simple. *Phys. Rev. Lett.* **77**, 3865-3868 (1996).
9. Setyawan, W.Curtarolo, S. High-throughput electronic band structure calculations: challenges and tools. *Comp. Mater. Sci.* **49**, 299-312 (2010).
10. Jang, J.G.Hong, J.-I. Alkyl chain engineering for enhancing the thermoelectric performance of single-walled carbon nanotubes–small organic molecule hybrid. *ACS Appl. Energy Mater.* **5**, 13871-13876 (2022).

11. Hsu, J.-H., Choi, W., Yang, G. & Yu, C. Origin of unusual thermoelectric transport behaviors in carbon nanotube filled polymer composites after solvent/acid treatments. *Org. Electron.* **45**, 182-189 (2017).
12. Kim, S.L., Choi, K., Tazebay, A. & Yu, C. Flexible power fabrics made of carbon nanotubes for harvesting thermoelectricity. *ACS Nano* **8**, 2377-2386 (2014).
13. Shi, X.-L., Zou, J. & Chen, Z.-G. Advanced thermoelectric design: from materials and structures to devices. *Chem. Rev.* **120**, 7399-7515 (2020).
14. Hada, M. et al. One-minute joule annealing enhances the thermoelectric properties of carbon nanotube yarns *via* the formation of graphene at the interface. *ACS Appl. Energy Mater.* **2**, 7700-7708 (2019).

REVIEWERS' COMMENTS

Reviewer #1 (Remarks to the Author):

Authors addressed well issues raised by reviewers, the revised manuscript is satisfactory and acceptable.

Reviewer #2 (Remarks to the Author):

The authors have addressed all my comments. It can be accepted now.

Reviewer #3 (Remarks to the Author):

The manuscript has been well revised, which can be accepted for publishing in NC.